# *CXCL1*: Gene, Promoter, Regulation of Expression, mRNA Stability, Regulation of Activity in the Intercellular Space

**DOI:** 10.3390/ijms23020792

**Published:** 2022-01-12

**Authors:** Jan Korbecki, Katarzyna Barczak, Izabela Gutowska, Dariusz Chlubek, Irena Baranowska-Bosiacka

**Affiliations:** 1Department of Biochemistry and Medical Chemistry, Pomeranian Medical University in Szczecin, Powstańców Wlkp. 72 Av., 70-111 Szczecin, Poland; jan.korbecki@onet.eu (J.K.); izabela.gutowska@pum.edu.pl (I.G.); d.chlubek@pum.edu.pl (D.C.); 2Department of Conservative Dentistry and Endodontics, Pomeranian Medical University in Szczecin, Powstańców Wlkp. 72 Av., 70-111 Szczecin, Poland; kasiabarczak@vp.pl

**Keywords:** *CXCL1*, MGSA, Gro-α, CXCR2, inflammation, cancer, tumor, chemokine, neutrophil

## Abstract

*CXCL1* is one of the most important chemokines, part of a group of chemotactic cytokines involved in the development of many inflammatory diseases. It activates CXCR2 and, at high levels, CXCR1. The expression of *CXCL1* is elevated in inflammatory reactions and also has important functions in physiology, including the induction of angiogenesis and recruitment of neutrophils. Due to a lack of reviews that precisely describe the regulation of *CXCL1* expression and function, in this paper, we present the mechanisms of *CXCL1* expression regulation with a special focus on cancer. We concentrate on the regulation of *CXCL1* expression through the regulation of *CXCL1* transcription and mRNA stability, including the involvement of NF-κB, p53, the effect of miRNAs and cytokines such as IFN-γ, IL-1β, IL-17, TGF-β and TNF-α. We also describe the mechanisms regulating *CXCL1* activity in the extracellular space, including proteolytic processing, *CXCL1* dimerization and the influence of the ACKR1/DARC receptor on *CXCL1* localization. Finally, we explain the role of *CXCL1* in cancer and possible therapeutic approaches directed against this chemokine.

## 1. Introduction

Intercellular signaling is an essential part of the functioning of a multicellular organism, involving the flow of information between different cells through either direct contact or via factors secreted outside the cell: simple compounds such as lactate [1], or large structures such as extracellular vesicles [2]. One group of factors responsible for intercellular signaling is chemokines, a group of about 50 chemotactic cytokines grouped according to the conserved *N*-terminal cysteine motif: CXC chemokines (α-chemokines), CC chemokines (β-chemokines), C chemokines (γ-chemokines) and CX_3_C chemokines (δ-chemokines) [3]. 

CXC motif chemokine ligand 1 (*CXCL1*) belongs to the sub-family of CXC chemokines [3,4]. Considering the number of available scientific papers, *CXCL1* is one of the four most studied CXC chemokines, with more than 5000 experimental papers available on PubMed (https://pubmed.ncbi.nlm.nih.gov; accessed date: 15 November 2021), with more than 1000 of these on the role of *CXCL1* in cancer. Although each year more than 400 new papers are published about this chemokine, there are no available review papers summarizing the current knowledge in this area. This paper is part of a greater project that aims to provide a comprehensive series of reviews on *CXCL1*; our focus is on the regulation of expression and activity of this chemokine in the intercellular space.

## 2. The Name ‘*CXCL1*’

*CXCL1* was first described in the 1980s as an auto-stimulatory melanoma mitogen, to which it owes one of its first names: melanoma growth-stimulatory activity (MGSA) [5]. Another historical name of *CXCL1* is neutrophil-activating peptide-3 (NAP-3), a term associated with its ability to induce neutrophil chemotaxis [6]. In subsequent years, *CXCL1* was shown to be a product of the growth-regulated gene (*Gro*) [7,8]. Since then, *CXCL1* has functioned under two names: MGSA and growth-regulated (or -related) oncogene (GRO) [8,9,10]. In the early 1990s, two more chemokines were identified that were very similar to GRO [11]. They were named GRO-β and GRO-γ, now known as *CXCL2* and *CXCL3* [12]. GRO itself was given the additional symbol α (GRO-α); another variant of this name is GRO-1 [9,11]. With the discovery of more CXC chemokines, a new and more systematic nomenclature was required. For this reason, a new classification was introduced based on the chemokine sub-family name and the number of the chemokine. Thus, GRO-α was named small inducible cytokine sub-family B, member 1 (SCYB1), and then *CXCL1* [12].

*CXCL1* has two important motifs in its sequence. One is the CXC motif (9–11 amino acids), which assigns this chemokine to CXC chemokines (or α-chemokines) [8]. The second is the ELR motif (6–8 amino acids), also located at the *N*-terminus, which makes *CXCL1* one of the ELR^+^ CXC chemokines, currently also known as CXC motif chemokine receptor 2 (CXCR2) receptor agonists [13,14].

## 3. CXCL1: Gene and Transcriptional Regulation

### 3.1. 4q12–q13 CXC Chemokine Gene Cluster

The *CXCL1* gene is localized in the 4q12–q13 CXC chemokine gene cluster [12,15]. This region includes the closely located loci of other CXC sub-family chemokines, including *CXCL2*/GRO-β, *CXCL3*/GRO-γ, *CXCL4*/PF-4, CXC motif chemokine ligand 4 variant-1 (*CXCL4L1*), *CXCL5*/ENA-78, *CXCL6*/GCP-2, *CXCL7*/PPBP/NAP-2, and *CXCL8*/IL-8 [4,16,17]. It also contains the CXC motif chemokine ligand 1 pseudogene 1 (*CXCL1P1*) previously described as a growth-regulated oncogene δ-pseudogene (GRO-δ) [18]. Near the cluster, at 4q21, are the ligands of CXCR3 and *CXCL13* [4,17]. Very often the 4q12–q13 gene cluster is amplified in cancer, particularly in Barrett neoplasia [19]. In breast cancer, 7.5% of primary tumors have a *CXCL1* gene amplification. The frequency of *CXCL1* amplification increases with disease progression [20]. The amplification of the *CXCL1* gene is observed in 20% cases of breast cancer metastasis [20]. The duplication of the 4q12–q13 CXC chemokine gene cluster is associated with a higher incidence of cancer, and is particularly found in individuals from melanoma-prone families [21].

The sequences of the chemokine genes in the 4q12–q13 CXC chemokine gene cluster are similar and located close to each other, which suggests that these genes originated as a result of duplication [4]. In general, all CXC chemokines from this gene cluster can be divided into two groups [3], *CXCL6* and *CXCL8* (which, at similar concentrations, activate both CXCR1 and CXCR2 receptors), and *CXCL1*, *CXCL2*, *CXCL3*, *CXCL5*, and *CXCL7* (which, at about 1 nM concentration, activate CXCR2; and at a concentration about 100 times higher, activate CXCR1) [3,22,23,24,25]. Each of the groups consists of chemokines that share the same properties and activate the same receptor at similar concentrations. Nevertheless, significant differences do occur in the action of individual chemokines, as the expression and secretion of each chemokine is sometimes separately regulated and depends on the cell type [26]. For this reason, to understand the interaction of all representatives of this gene cluster, it is necessary to understand the regulation of each CXC chemokine. 

At this point, it should be noted that the ancestor of today’s mammals (cretaceous period, separation of placental mammals from marsupials) had far fewer CXC chemokine genes. These genes were duplicated, and today, this gene cluster has seven CXC chemokine genes in mice and nine in humans [4]. The opossum has only three chemokine genes, and birds have between two and three depending on the species. While the accumulation of genetic changes in individual CXC chemokine genes occurred independently of each other, duplication may have occurred multiple times at different times. Therefore, mouse lipopolysaccharide-induced CXC chemokine (LIX) is as similar to rat LIX (a different species) as the human *CXCL5* is to human *CXCL6* (same species/genome) [27]. Therefore, a given human CXC chemokine cannot be unambiguously assigned to a mouse or rat CXC chemokine from this cluster. While it may be that in one disease or laboratory model a given chemokine in a human or mouse/rat is upregulated, when studying a second disease the expression of the given chemokine will be upregulated only in one species [28]. For this reason, the role of human *CXCL1* in a given disease cannot be compared with rat cytokine-induced neutrophil chemoattractant-1 (CINC-1) [29] or mouse keratinocyte-derived chemokine (KC) [30,31] without performing experiments on samples from human patients.

Importantly, given the arrangement of the genes on chromosomes, the human *CXCL1* gene does not correspond to the murine *KC* gene, but to the murine dendritic cell inflammatory protein-1 (*DCIP-1*) gene [4]. Even though the murine *KC* gene corresponds to the human *CXCL3*, *CXCL5* and *CXCL7* genes, in many papers, *CXCL1* and KC are used interchangeably. For this reason, in this review, we distinguish murine KC from human *CXCL1*.

### 3.2. CXCL1: Promoter

The *CXCL1* gene consists of four exons and three introns. The exons and introns have a total length of 1845 bp [32]. Preceding the transcription start site is a TATA box (at a locus from −30 to −24 bp) preceded by numerous sequences to which various transcriptional activators and transcriptional repressors are attached.

Due to the significance of *CXCL1* in inflammatory responses, the most important method for the induction of expression of this chemokine is the activation of NF-κB. At a locus −78 to −66 bp is the nuclear factor κB (NF-κB) binding site [33,34,35,36]. The NF-κB p50/p65 heterodimer attaches to it, inducing *CXCL1* expression [37]. Therefore, through NF-κB activation, *CXCL1* expression is increased by cytokines such as interleukin-1β (IL-1β) [37,38], tumor necrosis factor-α (TNF-α) [37,38] and interleukin-17 (IL-17) [39,40,41]. Importantly, the activation of NF-κB by the IL-17 receptor (IL-17R) occurs via the NF-κB activator 1 (Act1) → TNF receptor-associated factor 6 (TRAF6) → transforming growth factor β (TGF-β)-activated kinase 1 (TAK1) pathway [39,40]. The main effect of IL-17 on *CXCL1* expression is an increase in the stability of *CXCL1* mRNA [42,43]. In cancer cells, basal NF-κB activation is high, which results in high basal *CXCL1* expression [35]. Therefore, in cancer cells, pro-inflammatory cytokines such as IL-1β and TNF-α can only increase the stability of *CXCL1* mRNA [37]. *CXCL1* can also increase its own expression via NF-κB activation, a mechanism that is important in cancer [8,44,45]. The effect of pro-inflammatory cytokines on *CXCL1* expression is important because of inflammatory responses in malignant tumors. Depending on the type of cancer, there is an upregulation in pro-inflammatory cytokines, e.g., TNF-α [46] and IL-17 [47,48], which are involved in tumorigenesis.

In some cases, NF-κB can decrease *CXCL1* expression. In hepatocytes, there is an important NF-κB p50/p50 homodimer that binds to the *CXCL1* promoter to recruit the co-repressor histone deacetylase 1 (HDAC1) [49]. This results in reduced *CXCL1* expression, which is important for preventing chronic liver disease.

The NF-κB binding site also contains the AT motif (the exact locus is −74 bp and −73 bp), which can attach a high-mobility group, AT-hook 1 (HMGA1) (previous name: high-mobility group-I(Y) (HMGI(Y))) [34,50]. This protein is important for the full activation of the *CXCL1* promoter [34].

In the induction of *CXCL1* expression by the exposure of cells to TNF-α, the formation of an NF-κB complex with cut-like homeobox 1 (CUX1) is important, and is much more intensely induced in the simultaneous exposure to IL-17 and TNF-α [51]. This complex attaches to the *CXCL1* gene promoter at the NF-κB binding site in the −94 bp to −84 bp region of the *CXCL1* promoter, i.e., where the binding site for CUX1 is located, leading to an increase in *CXCL1* expression. This process is significant in rheumatoid arthritis, which is marked by high concentrations of IL-17 and TNF-α at disease sites [51].

The *CXCL1* gene promoter also contains the immediate upstream region (IUR) at loci −93 bp to −78 bp (Figure 1) [50,52]. This region is directly upstream of the NF-κB binding site [33,34,35,36] and can attach to the human CUT homeodomain protein/CCAAT displacement protein (CDP) [53]. Then, CDP disrupts the interaction with the NF-κB of CREB-binding protein (CBP) or p300/CBP-association factor (PCAF) [52]. As CBP and PCAF are coactivators important for the function of NF-κB [54], CDP decreases the expression of *CXCL1*, which depends on NF-κB [52,53]. Poly(ADP-ribose) polymerase (PARP1) can also attach to the IUR [55,56], although only in the inactive state of PARP1. This inhibits the binding of NF-κB to the *CXCL1* promoter, and thus the expression of this gene. In contrast, activated PARP1 loses its ability to bind to the IUR, which results in NF-κB binding to the *CXCL1* promoter and an increase in *CXCL1* expression [56]. The involvement of PARP1 is important in melanoma tumorigenesis; in normal melanocytes, PARP1 is inactive and inhibits *CXCL1* expression, while in melanomas PARP1 is active.

Locus −129 bp to −119 bp is the specificity protein 1 (Sp1) and specificity protein 3 (Sp3) binding site [34,50], important in the basal expression of *CXCL1* and the induction of *CXCL1* expression by IL-17 and TNF-α [41,57]. This site is also crucial for regulation of the expression of *CXCL1* by the interferon-γ (IFN-γ) signal transducer, and the activator of transcription (STAT)1 can also inhibit *CXCL1* expression. In particular, in peritoneal mesothelial cells, upon exposure of these cells to IFN-γ, STAT1 binds to the −154 bp region upstream of the transcription start site of the *CXCL1* gene, which results in a decreased expression of *CXCL1* [57] and reduced binding of Sp1 to the *CXCL1* promoter. STAT1 may also increase the expression of *CXCL1*. However, its effect will depend on the selected cell model and the factor acting on these cells. In pancreatic ductal adenocarcinoma cells, there is an induction of *CXCL1* expression by IL-35 [58], which is associated with the direct binding of the STAT1/STAT4 heterodimer to the *CXCL1* promoter.

#### 3.2.1. The Regulation of CXCL1 Expression by TGF-β and HGF

TGF-β reduces *CXCL1* expression [59,60]. At locus −1247 bp, upstream of the transcription start site of the *CXCL1* gene, is the TGF-β-inhibitory element (TIE), and at locus −560 bp is the SMAD-binding element (SBE) [61]. SMAD family member 4 (SMAD4) is the main factor in the signaling from the TGF-β receptor, but does not bind to these sites. Therefore, the effect of TGF-β on *CXCL1* expression is indirect, likely reducing the activity of either NF-κB or other signaling pathways [61,62].

In contrast, in murine KC, a paralog for human *CXCL1*, there is a different mechanism by which TGF-β regulates the expression of this chemokine. At loci −249 bp to −246 bp and −144 bp to −141 bp, upstream of the transcription start site of the *KC* gene, are SBEs, which can bind SMAD2/3, reducing KC expression [63].

At the upstream transcription start site of the *KC* gene at loci −128 bp to −120 bp is the CCAAT/enhancer binding protein-β (C/EBP-β) binding motif. Upon activation of c-Met by the hepatocyte growth factor (HGF), C/EBP-β is activated and attaches to the C/EBP-β binding motif on the *KC* promoter, thus increasing the expression of this murine KC chemokine. Importantly, this mechanism only occurs in murine cells [63]. Based on the database “https://www.ncbi.nlm.nih.gov/gene”; accessed date: 15 November 2021, we did not find any C/EBP-β binding motif according to the cited paper on mouse KC, i.e., 5′-TGGAGCAAG-3′ or any sequence complementary to it up to −2 kbp upstream of the human *CXCL1* gene promoter. Therefore, theoretically, C/EBP-β does not directly affect the expression of human *CXCL1*. In addition, studies on human cells do not show that TGF-β and SMADs directly affect human *CXCL1* expression [61]. Nevertheless, further studies on the mechanisms of regulation of *CXCL1* expression by HGF and C/EBP-β in humans are required.

The regulation of *CXCL1* expression by TGF-β is an important mechanism observed in cancer [61,63]. For example, SMAD4 expression is reduced in colorectal cancer cells, as it interferes with the action of TGF-β, and thus leads to an increase in *CXCL1* expression in the cancer cell [61]. Notably, in a prostate cancer tumor, in cancer-associated fibroblasts (CAF), there is a decrease in TGF-β type II receptor (TβRII) expression [64]. This decreases the effect of TGF-β on these cells, and thus leads to the increased expression of *CXCL1* in these cells. As *CXCL1* is a chemotactic factor for neutrophils [65], an increase in *CXCL1* expression in colorectal cancer tumors results in the recruitment of tumor-associated neutrophils (TAN), cells with pro-cancer properties [61,66]. Additionally, *CXCL1* causes tumor cell migration, and consequently, metastasis [60].

#### 3.2.2. Significance of p53 Transcription Factor Family in *CXCL1* Expression

Gain-of-function and loss-of-function mutations of the *TP53* gene are very common in cancer cells and lead to changes in the expression of various genes [67,68]. Loss-of-function *TP53* mutations cause an increase in *CXCL1* expression [69] due to reduced p53-induced inhibition of NF-κB activation [70,71]. If p53 loses its functions, there is an increase in NF-κB activity and in the expression of genes dependent on this transcription factor. This mechanism is not universal in all cells because in monocytes and macrophages, p53 together with NF-κB increase the expression of pro-inflammatory cytokines [72]. In addition, the interaction of *CXCL1* with p53 is complicated, as CXCR2 activation reduces p53 expression, this action is associated with protein kinase B (PKB)/Akt activation, which in turn activates murine double minute 2 (Mdm2) [73].

The gain-of-function *TP53* mutations cause mutated p53 proteins to directly bind to the *CXCL1* promoter, which results in an increased *CXCL1* expression in the tumor cell [74,75]. The exact mechanism of the increase in *CXCL1* expression depends on the type of p53 mutation. DNA-contact p53 mutants (for example R248Q and R273H) bind directly to the *CXCL1* promoter [75]. Some of the single-nucleotide polymorphisms (SNP) of p53 may increase the ability of p53 with the aforementioned mutation to attach to the *CXCL1* promoter. An example of this is the P72R SNP polymorphism of p53 (*rs1042522*) [75]. The R72 p53 mutant has a higher binding capacity to the *CXCL1* promoter than the P72 p53 mutant. DNA-contact p53 mutants also increase NF-κB activation, which increases the expression of genes dependent on this transcription factor, including *CXCL1* [69].

Another type of mutation is the Zn^2+^ region conformational p53 mutant. Examples of such p53 mutants are R175H and H179R, which attach directly to the *CXCL1* promoter, and thus increase *CXCL1* expression [74]. It appears that p53 with this type of mutation also increases *CXCL1* expression by increasing H-Ras activity [69]. This effect is related to the interaction of mutant p53 with B-cell translocation gene 2 (BTG2). Significantly, NF-κB is not required in the induction of *CXCL1* expression by Zn^2+^ region conformational p53 mutants [74].

The last type of mutation is the L3 loop conformational p53 mutant [69]. An example of such a mutation is G245S p53. Although this type of p53 mutation increases *CXCL1* expression, the effect is much smaller than when p53 expression is significantly reduced [69]. This is likely due to the reduced effect of p53 with such a mutation, relative to wild-type (WT) p53.

Another protein that attaches near the *CXCL1* gene is p63, a transcription factor from the p53 transcription factor family [76,77]. Both proteins (p53 and p63) induce the expression of different genes, but can also cooperate in the expression of the same genes. p63 attaches to the −3 kb region of the upstream transcription start site of the *CXCL1* gene, thus increasing its expression [77]. This transcription factor may also cooperate with NF-κB in increasing *CXCL1* expression. The significance of p63 in *CXCL1* expression was found in tumors, particularly in pancreatic ductal adenocarcinoma cells.

#### 3.2.3. *CXCL1* Expression and Hypoxia

The promoter of murine KC, a paralog for human *CXCL1*, contains five hypoxia response element (HRE) sequences at loci −1309 bp, −433 bp, −313 bp, −302 bp and −289 bp [78]. The expression of this gene is induced by hypoxia-inducible factor (HIF)-2, as shown by studies in mouse epithelial cells [79] and murine chondrocytes [78]. HIF-1 also increases KC expression in murine myeloid-derived suppressor cells (MDSCs) [80]. *CXCL1* expression may depend indirectly on HIF-1; for example, in human aortic endothelial cells and mouse aortic endothelial cells, HIF-1 increases miR-19a expression [81], which indirectly increases the expression of *CXCL1*. Based on “https://www.ncbi.nlm.nih.gov/gene; accessed date: 15 November 2021”, we identified seven sequences thought to be an HRE, i.e., 5′-A/GCGTG-3′ and complementary ones in the region up to the −2 kbp upstream transcription start site of the human *CXCL1* gene [82,83]. HRE is the binding site of HIF-1 and HIF-2, transcription factors activated by hypoxia. Therefore, it seems theoretically possible that hypoxia increases *CXCL1* expression—this was already shown by a study involving hepatocellular carcinoma cells [84]. Nevertheless, this effect may depend on the particular research model, as hypoxia did not affect *CXCL1* expression in lung adenocarcinoma cells [85].

#### 3.2.4. Other Mechanisms That Alter CXCL1 Promoter Activity

In addition to the effects on *CXCL1* expression of the aforementioned factor binding sites near the transcription start site, there are also regulatory sequences located at various distances from the *CXCL1* gene.

*CXCL1* expression is increased in an AP-1-dependent manner, as shown by experiments using IL-17 [41] and TNF-α [86,87].

The transcription factors responsible for the induction of *CXCL1* expression by TNF-α also include early growth response gene 1 (Egr-1) [50], which binds directly to the *CXCL1* promoter at two loci: −367 bp and −134 bp. This process is significant in cancer, particularly in esophageal cancer, where Erg-1 is frequently overexpressed [88].

Studies on breast cancer cells showed that a breast cancer susceptibility gene 1 (BRCA1) complex with GATA-binding protein 3 (GATA3) reduced *CXCL1* expression [89]. Nevertheless, further studies are required to determine whether this effect is direct or indirect.

Other transcription factors also attach to the *CXCL1* promoter. In particular, the activation of transcription factor 2 (ATF2) and acute myeloid leukemia 1 (AML1) is responsible for the increase in *CXCL1* transcription by IL-17, which attaches to the *CXCL1* promoter [41].

At locus −277 bp, upstream of the transcription start site of the *CXCL1* gene, a binding site for myeloid ecotropic viral integration site 1 (MEIS1) was identified [90]. MEIS1 is a transcription factor that is overexpressed in cancer cells, particularly in ovarian cancer, increasing the expression of many chemokines including CCL18, CCL4, CXCL7, and indirectly increasing the expression of *CXCL1*, as the identified binding site of this transcription factor appears to be non-functional.

At −375 bp upstream of the transcription start site of the *CXCL1* gene, a binding site for the microphthalmia-associated transcription factor (MITF) was identified [91]. This is a transcription factor important for melanocyte differentiation. It also undergoes overexpression in melanoma. Therefore, the increase in *CXCL1* expression in melanoma depends partly on MITF.

The −551 bp to −517 bp region upstream of the transcription start site region of the KC gene contains a binding site for Y-box protein-1 (YB-1) [92]. KC expression is induced upon the binding of this protein; this process is relevant in a murine bile duct ligation model, where the expression of KC increases in the liver in a YB-1-dependent manner. However, further studies are required to confirm the role of YB-1 in humans.

The −984 bp to −301 bp region, upstream of the transcription start site of the *CXCL1* gene, contains a binding site for Snail, a transcription factor crucial for epithelial–mesenchymal transition (EMT) [93,94], a process important in cancer cell migration, invasion and metastasis. After EMT, the tumor cell begins to migrate. The increased expression of *CXCL1* in this cell plays an important role in migration [95] and metastasis formation [64,96] by stimulating the tumor cell to migrate, and causing the supporting adhesion of the tumor cell to target tissues, as well as in the recruitment of different cells in metastasis.

Further from the transcription start site of the *CXCL1* gene at the locus −2 kbp, a Hey-like (HeyL) binding site is located [97,98]. This factor belongs to Notch signaling, which is overexpressed in many cancers, such as in breast cancer. Because *CXCL1* is an angiogenic chemokine [13,14], Notch signaling in cancer increases *CXCL1* expression, and thus causes angiogenesis.

Some proteins regulating the expression of *CXCL1* can be bound very far from the *CXCL1* gene. In particular, avian v-maf musculoaponeurotic fibrosarcoma oncogene homolog (MAF)F binds to three sites, precisely at the −15 kpb, −12.5 kpb and −7.5 kbp upstream of the transcription start site of *CXCL1* gene [99]. MAFF increases *CXCL1* expression in human term myometrium [99,100]. However, to date, the function of *CXCL1* during labor is unknown. The mechanism of induction of *CXCL1* expression in human term myometrium is not related to inflammatory factors, which means that *CXCL1* expression is induced by a specific transcription factor independent of inflammatory responses [99]. This shows that *CXCL1* is not only a mediator of inflammatory responses but may also have its own physiological functions unrelated to inflammation.

Histone methylation is another important means of regulating *CXCL1* expression. A −2.0 to −1.5 kbp fragment upstream of the transcription start site of the *CXCL1* gene undergoes histone H3 Lys36 trimethylation (H3K36me3) by histone H3 lysine 36 methyltransferase SET-domain-containing 2 (SETD2) [101]. This leads to a reduction in *CXCL1* expression. This process is important because, in cancers such as breast cancer [102], glioblastoma multiforme [103], hepatocellular carcinoma [104] and lung adenocarcinoma [101] the *SETD2* gene is either mutated or there is a decreased expression of the product of this gene. However, in castration-resistant prostate cancer, SETD2 is an oncogene that supports tumor growth [105]. The precise regulation of *CXCL1* expression by SETD2 is not known. As this enzyme modifies a fragment very far from the *CXCL1* gene, it is possible that it alters the ability to bind some proteins that are important for the regulation of gene expression (Table 1).

### 3.3. Regulation during Transcription

The induction of signaling pathways and attachment of all transcription factors to the promoter of the *CXCL1* gene is followed by transcription. RNA polymerase II (Pol II) begins to transcribe the *CXCL1* mRNA fragment with a length of approximately 50 nucleotides [106]. Then, transcription elongation requires the phosphorylation on Ser^2^ of Pol II by positive transcription elongation factor b (P-TEFb). Hairy and Enhancer of Split-1 (Hes1) prevent this phosphorylation, and therefore inhibit *CXCL1* transcription [106]. This regulation occurs particularly in macrophages treated with pro-inflammatory agents such as lipopolysaccharide (LPS) [107].

## 4. *CXCL1* mRNA Stability as a Method to Regulate *CXCL1* Expression

The production of a 1.1–1.2 kb-long transcript [8] is followed by the next step: the regulation of *CXCL1* expression, associated with changes in the stability of *CXCL1* mRNA [108]. *CXCL1* mRNA is a transcript with a low half-life; if the cell is not influenced by any factors that increase its stability, then this transcript is degraded within 1 to 4 h [109,110]. The half-life of *CXCL1* mRNA is estimated to be approximately 15 min [37]. After this time, from the ends of the 3′-untranslated region (UTR) of *CXCL1* mRNA, an approximately 130-nucleotide fragment is removed [109,110]. *CXCL1* mRNA is transformed into a 0.9 kb transcript. However, there are many mechanisms, some of which increase and some of which decrease the half-life of *CXCL1* mRNA. In particular, regions 6–23 and 632–651 on the 3′-UTR of *CXCL1* mRNA are responsible for increasing the stability of the transcript. Region 562–581 on the 3′-UTR of *CXCL1* mRNA is responsible for decreasing the stability of *CXCL1* mRNA [111]. The AUUUA motifs on the 3′-UTR also act to reduce the stability of *CXCL1* mRNA [42].

Various proteins that are activated by factors such as IL-1 [37,109,110], TNF-α [37] and IL-17 [43] are responsible for increasing *CXCL1* mRNA stability. This effect depends on the cell type and tissue selected as the research model, as these factors can also increase *CXCL1* transcription, e.g., depending on NF-κB activation [37,39,57]. Notably, in non-cancer cells, where there is no high basal NF-κB activation, transcriptional regulation plays a major role in regulating *CXCL1* expression [37]. Additionally, no less important for *CXCL1* mRNA stability are miRNAs, which cause the decay of this transcript [112,113,114,115,116,117]. All of the factors mentioned in this section lead to the degradation or improvement in the stability of *CXCL1* mRNA. This alters the half-life of *CXCL1* mRNA, and consequently, the number of these transcripts in the cytoplasm, at unchanged *CXCL1* gene transcription levels.

### 4.1. Role of Cytokines in Regulating CXCL1 Expression by Altering mRNA Stability. The Mechanisms of IL-17-Induced Effects on CXCL1 Expression

*CXCL1* mRNA decay is IL-17-sensitive. In unstimulated cells, splicing factor 2 (SF2)/alternative splicing factor (ASF) attaches to the 3′-UTR of *CXCL1* mRNA, which reduces the stability of this transcript [118]. This factor attaches to *CXCL1* mRNA sites other than tristetraprolin (TTP). The activation of the receptor for IL-17 leads to activation of Act1, TRAF2 and TRAF5. Act1 causes the phosphorylation of SF2/ASF via the inhibitor of NF-κB kinase ε (IKKε) [43] and forms the Act1-TRAF2-TRAF5-SF2/ASF complex [118,119]. In this complex, SF2/ASF is no longer bound to *CXCL1* mRNA, which increases the stability of this transcript. At the same time, Act1 causes K^63^-linked polyubiquitination of the human antigen R (HuR) [108]. This process is dependent on Ubc13-Uev1A E2 complex. Therefore, modified HuR binds to the AU-rich element (ARE) on *CXCL1* mRNA, causing its stabilization; HuR also promotes the translation of *CXCL1*.

Act1 can also cause the phosphorylation of decapping 1 (Dcp1) via TANK-binding kinase 1 (TBK1) [43]. This inhibits the decapping of *CXCL1* mRNA, and of other transcripts. On the 3′-UTR of *CXCL1* mRNA, there is a fragment at locus 780–900 that forms a secondary structure with four stem loops. This is a “similar expression to fibroblast growth factor genes + IL-17R” (SEFIR)-binding element. Act1 binds directly to it, which increases the stability of the transcript in question by competing with SF2/ASF binding [43].

IL-17 increases *CXCL1* mRNA stability, and thus *CXCL1* expression. Nevertheless, IL-17 can also increase *CXCL1* transcription, but this effect is not as significant as the change in *CXCL1* mRNA stability and may also depend on the research model chosen [41]. An increase in *CXCL1* transcription is associated with NF-κB activation by the IL-17R receptor [39,40,41]. Additionally, other transcription factors increase *CXCL1* expression, including AP1, ATF2, AML1 and SP1 [41,57].

In macrophages [120], keratinocytes [36] and fibroblasts [121], tristetraprolin (TTP) is responsible for reducing the stability of KC mRNA, as shown by experiments on mouse cells. TTP binds to AREs and, more specifically, to AUUUA motifs on the 3′-UTR of KC mRNA. This leads to a recruitment of the deadenylases, and consequently, to a decay of KC mRNA. However, activated p38 MAPK by LPS can activate the MAPK-activated protein kinase 2 (MK2) [122]. This kinase phosphorylates TTP, which does not affect the binding of TTP to mRNA but prevents the recruitment of deadenylases by this factor and prevents the decay of KC mRNA [123]. However, research on human cells showed that TTP does not affect the stability of *CXCL1* mRNA [42].

### 4.2. The Role of miRNAs in the Regulation of CXCL1 Expression

To date, many miRNAs have been found to be involved in reducing *CXCL1* expression. In particular, many of them have important roles in cancer due to the pro-tumorigenic properties of *CXCL1* (Figure 2) [5,114,124,125]. The development of cancer is associated with a decrease in the expression of miRNAs regulating *CXCL1* expression in the tumor, and consequently, is associated with an increase in the expression of this chemokine. In ovarian cancer, renal cancer [112] and hepatocellular carcinoma [114], there is a downregulation of the miR-200 family [112]; in ovarian cancer there is a downregulation of miR-27b-5p [116]; and in gastric cancer, there is a downregulation of miR-204 [115]. Other examples include miR-302e in colorectal cancer [117] and miR-141 in non-small cell lung cancer [113].

A change in miRNA expression may also account for the effects of some anticancer substances. An example of this is the increased expression of miR-181b in breast cancer cells by curcumin [126]. This miRNA directly decreases *CXCL1* expression, which is one of the anticancer mechanisms of curcumin.

It is also possible that miRNAs affect signaling pathways that increase the expression of chemokines, which would mean that such miRNAs indirectly decrease *CXCL1* expression. An example of this is miR-155 in tumor-infiltrating MDSCs [80]. This miRNA decreases HIF-1 levels, and thus the expression of genes dependent on the transcription factor HIF-1, such as *CXCL1*.

*CXCL1* expression can also be indirectly increased by miRNAs. Examples of this are miR-155, miR-193b and miR-210 [127]. These miRNAs are secreted in extracellular vesicles by cancer cells, and then increase *CXCL1* expression in fibroblasts. This mechanism was demonstrated in gastric cancer [127].

It is not only in cancers that the regulation of *CXCL1* expression by miRNAs occurs. During cerebral ischemia–reperfusion injury, there is a downregulation of miR-429 in brain microvascular endothelial cells [128] and miR-532-5p in brain tissues [129]. At least for miR-532-5p, this was shown to be related to the promoter of this miRNA: hypermethylation. Both miRNAs directly reduce the expression of *CXCL1*. Therefore, when the levels of these miRNAs are decreased, there is an increase in *CXCL1* expression, which leads to brain tissue damage during cerebral ischemia–reperfusion injury.

Additionally, studies of extracellular vesicles from murine, adipose-derived mesenchymal stem cells showed that they contain miR-150-5p [130]. This miRNA reduces the expression of KC. This chemokine is important in the development of hepatic fibrosis. Therefore, adipose-derived mesenchymal stem cells may inhibit the development of hepatic fibrosis.

Another miRNA that reduces *CXCL1* expression is miR-7641 [131]. The expression of this miRNA occurs in human embryonic stem cells and decreases during endothelial cell biogenesis. This increases the expression of *CXCL1*, a chemokine also important in angiogenesis.

## 5. *CXCL1*: From Translation to Extracellular Factor

After translation, the synthesized *CXCL1* precursor is 107aa long [132]. A signal peptide is removed from its *N*-terminus, which shortens the precursor to 73aa. Two other amino acids can also be removed from the C-terminus. In addition, two disulfide bridges are formed from all four cysteine residues in *CXCL1* (Figure 3) [8,133,134]. The disulfide bridges give the appropriate structure to *CXCL1*, which determines the properties of this chemokine [135]. Apart from the aforementioned modifications, this chemokine does not undergo other post-translational modifications such as glycosylation, sulfation or phosphorylation [132].

Following the production of *CXCL1* in the cytoplasm, this chemokine is sorted to vesicles [139]. It was shown that, at least in endothelial cells, these are histamine-responsive intracellular compartments located throughout the cytoplasm [140]. *CXCL1* is not sorted by the Weibel–Palade body (WPB), i.e., it occurs in different locations than *CXCL8* [140]. The localization of *CXCL1* allows for a rapid release of *CXCL1* under the influence of pro-inflammatory factors. An example of this is TNF-α, which causes a p38 MAPK- and PI3K-dependent release of *CXCL1* [86]. Additionally, in endothelial cells, IL-1β stimulation results in the sorting of serglycin and *CXCL1* into the same vesicles, and then a simultaneous secretion of both biomolecules [139].

## 6. *CXCL1* as an Extracellular Factor

### 6.1. CXCL1 and Glycosaminoglycans

Following its secretion outside the cell, *CXCL1* binds to glycosaminoglycans (GAGs), in particular heparan sulfate, chondroitin sulfate, and dermatan sulfate [141]. This mechanism is also important in the removal of excess *CXCL1*, which inhibits an overly intense pro-inflammatory response [142,143]. *CXCL1* attached to GAG can be readily released by enzymes that degrade GAG. An example of such an enzyme is the matrix metalloproteinase (MMP)7/matrilysin during colon injury in mice [143,144,145]. This enzyme causes the shedding of syndecan-1, which results in the release of syndecan-1/*CXCL1* complexes. This complex is responsible for the neutrophil influx to pro-inflammatory response sites. In addition, GAGs are important for the full activation of CXCR2 receptor by *CXCL1* [146]. Without GAGs, *CXCL1* binds poorly to the aforementioned receptor.

### 6.2. Proteolytic Processing as One of the Mechanisms Regulating CXCL1 Activity

After secretion, *CXCL1* undergoes further proteolytic processing, which regulates the activity of this chemokine. From the *N*-terminus, three, four or five amino acids are removed, which produce *CXCL1*(4–73), *CXCL1*(5–73), and *CXCL1*(6–73), respectively [147]. This increases *CXCL1* activity 30 times, as measured by its ability to induce the chemotaxis of treated cells. The process of amino acid removal may involve cathepsins, in particular cathepsin K, cathepsin L, and cathepsin S [148]. These proteases show their activity at a neutral pH and remove four amino acids from the *N*-terminus of *CXCL1*, resulting in the formation of *CXCL1*(5–73). Nevertheless, no proteases responsible for the formation of *CXCL1*(4–73) and *CXCL1*(6–73) have been identified so far. The identification of proteases responsible for processing *CXCL1* would allow new pathways regulating the activity of this chemokine to be demonstrated. However, studies on metalloproteinases, MMP1 and MMP9, showed that they do not exhibit any activity against *CXCL1* [149]. Instead, it was shown that *CXCL1* could be sliced inside the ELR sequence by MMP12, which resulted in the formation of *CXCL1*(7–73) and the inactivation of *CXCL1*.

### 6.3. Dimerization of CXCL1

*CXCL1* can be dimerized [150] and tends to form oligomers [132,151], especially at high concentrations. At concentrations from around 5 μM [152] to 20 μM [153], depending on the literature cited, half of the *CXCL1* molecules occur in the form of dimers. However, for mouse KC, which is the paralog of human *CXCL1*, this concentration can be higher: 36 μM [154]. This concentration is more than 100 times higher than the concentration of *CXCL1* in normal- and cancer-tissue homogenates [155,156]. At high concentrations, *CXCL1* proteins are dimerized during biochemical isolation [132,151]. Therefore, in living organisms, dimers can only exist locally. Surface GAG binds to various chemokines, causing them to aggregate [157]. Importantly, surface GAG binds *CXCL1* homodimers, for which it has a higher affinity than for the monomers of this chemokine [154,158]. That is, at low concentrations, *CXCL1* homodimers are bound to surface glycosaminoglycans, and the monomers are soluble forms of *CXCL1*. However, the arrangement of GAG-bound *CXCL1* prevents the activation of CXCR2 [158]. The activation of this receptor must be carried out by soluble *CXCL1*. GAGs, particularly heparan sulfate proteoglycans, appear to play an important role in CXCR2 receptor activation by *CXCL1* [146].

Both *CXCL1* monomers and homodimers have a similar ability to activate CXCR2 [153,159]. It appears that the synergistic action of both *CXCL1* monomers and homodimers is required for full *CXCL1* activity [160,161]. Therefore, at low concentrations, unmodified *CXCL1* has a greater activity than stable *CXCL1* homodimers and a greater activity than modified *CXCL1* monomers, so that they do not dimerize [159,160,161]. In contrast, at very high concentrations, both forms have similar properties because unmodified *CXCL1* undergoes homodimerization.

The mixture of monomers and homodimers of *CXCL1* is only a simplified model, as *CXCL1* can also form heterodimers with other CXC chemokines. In particular with *CXCL4* [150], *CXCL7* [162] and *CXCL8* [150]. Such a heterodimer binds to GAGs in a different arrangement than the *CXCL1* homodimer [162,163]. However, heterodimers appear to have the same activity in CXCR2 activation as the monomers of individual chemokines [162].

The formation of *CXCL1* heterodimers with other CXC chemokines has been poorly studied. Usually, the role of only a single selected chemokine is studied in a given process, even though the demonstration of interactions between different CXC chemokines may provide a better understanding of therapeutic methods against a selected chemokine, particularly in cancers, associated with the increased expressions of many various CXC chemokines, particularly in colorectal cancer [164], head and neck squamous cell carcinoma [165] and skin cutaneous melanoma [166].

### 6.4. CXCL1 Receptors

To date, three *CXCL1* receptors have been discovered—CXCR1, CXCR2 and atypical chemokine receptor 1 (ACKR1) [167]. Of these, *CXCL1* binds with high affinity to just two of them—CXCR2 and ACKR1. ACKR1 binds *CXCL1* with a dissociation constant (*K*_d_) of 1.81 nM [167], similar to the binding parameters of *CXCL1* to the CXCR2 receptor. Depending on the study, the half maximal effective concentration (EC_50_) for CXCR2 and 5.0 ± 1.7 nM [24] or 0.4 nM [168], with *K*_d_ of 3 nM [23].

*CXCL1* can bind to and activate CXCR1, but the binding parameters to this receptor are 300 to 1000 times weaker than for CXCR2 [23]. The EC_50_ for *CXCL1* to activate CXCR1 is 65 ± 13 nM [24] or 220 nM [168], while *K*_d_ is 300 nM [23] or 450–880 nM [169], depending on the study. The properties of *CXCL1* seem to mainly associate with CXCR2, especially in the light of its much higher affinity to *CXCL1* than that observed for CXCR1, even though some papers show that CXCR1 in some models is important for the action of *CXCL1* [170,171].

### 6.5. ACKR1 and CXCL1

ACKR1 was first described as a Duffy antigen, which forms the Duffy blood-group system, hence its original name: Duffy antigen receptor for chemokines (DARC) [172]. With a molecular mass of 47 kDa [173], ACKR1 is expressed on erythrocytes, and in the absence of expression in these cells, it is found in the capillaries and postcapillary venular endothelial cells [174,175]. ACKR1 is also an important receptor for malaria parasites, *Plasmodium vivax* and *Plasmodium knowlesi*, to infect erythrocytes [172,176,177]. For this reason, in tropical areas where these parasites occur, selection pressure results in the prevalence of ACKR1 alleles whose expression is not found on erythrocytes [172].

ACKR1 is not only a receptor for *CXCL1* but also for *CXCL8* and CC chemokines, CCL2 and CCL5, although not for CCL3 or CXCR3 receptor agonists [167,178]. The ACKR1 sequence does not contain a DRY motif [179], a motif associated with G-protein coupling [180,181]. Therefore, G-protein-dependent signal transduction does not occur after ligand binding. However, ACKR1 can activate extracellular signal-regulated kinase (ERK) mitogen-activated protein kinase (MAPK), which interfere with CXCR2 receptor function [182].

ACKR1 has an important role in chemokine action. On erythrocytes, it binds chemokines (including *CXCL1*) from the blood, which leads to the maintenance of *CXCL1* levels and regulation of the intensity of the inflammatory response [173,183,184].

In addition, on endothelial cells, ACKR1 is involved in immune cell migration. After it binds chemokines, it is endocytosed with the bound chemokine via a macropinocytosis-like process, dependent on cholesterol and dynamin II [185]. Significantly, the chemokine is not degraded in this process but translocated across the endothelium, which is followed by the migration of the immune cell through the endothelium. [175,185].

ACKR1 activation can also cause the signal transduction of ERK MAPK, which interferes with the function of CXCR2 on the same cell as ACKR1 [182]. This process leads to the inhibition of CXCR2-dependent cell migration, e.g., airway smooth muscle cells [182]. The interaction between ACKR1 and CXCR2 is also important in cancer. The high expression of ACKR1, associated with the inhibition of CXCR2, results in a better prognosis for patients with pancreatic ductal adenocarcinoma [186].

## 7. Significance of *CXCL1* in Tumors

*CXCL1* expression in cancer tumors is upregulated by numerous mechanisms at almost all possible regulatory steps: gene amplification [19], the activation of transcription by high basal NF-κB activation [35], effects of pro-inflammatory cytokines on *CXCL1* transcription and *CXCL1* mRNA stability [46,47,48], as well as miRNAs involved in regulating *CXCL1* expression in the tumor cell [112,114].

After *CXCL1* expression is induced by carcinogens, it participates in inflammatory responses by recruiting neutrophils. This leads to chronic inflammation, which causes tumor formation as demonstrated in the models of carcinogen-induced skin cancer [187,188]. *CXCL1* is produced in the tumor by cancer cells [189,190], and also by CAFs [189,191,192,193], MDSCs [80], mesenchymal stem cells (MSCs) [194] and tumor-associated macrophages (TAMs) [189,195,196]. Since the first experiments on *CXCL1*, this chemokine is considered an important factor in cancer development. It was first classified as a factor produced by cancer cells that increased their proliferation [5], hence its original names: “growth regulated oncogene” and “melanoma growth stimulatory activity factor” [5,7,8]. Over time, other pro-tumor properties of this chemokine were discovered. In addition to increasing proliferation [5,197], *CXCL1* also induces cancer cell migration, particularly EMT [193,196,197]. Produced by lymphatic endothelial cells (LECs), *CXCL1* enables tumor cell migration into the lymphatic vessels during lymphangiogenesis, leading to lymph node metastasis.

*CXCL1* is a chemotactic factor for neutrophils [65]. Additionally, it causes the mobilization of these cells from the bone marrow [198]. These processes lead to the recruitment of TAN [199,200] and granulocytic, myeloid-derived suppressor cells (G-MDSC) [195,201,202] into the tumor niche. In doing so, this chemokine also increases the expansion of monocytic, myeloid-derived suppressor cells (Mo-MDSCs) in the bone marrow, which increases the number of these cells in the body, and thus in the tumor after the recruitment of these cells by other chemokines [203]. *CXCL1* can also induce recruitment of regulatory T cells (T_reg_) [113] and MSCs [204] into the tumor niche.

Another no-less-important property of *CXCL1* is its ability to induce angiogenesis [13,14]. In particular, the expression of *CXCL1* is increased by the vascular endothelial growth factor (VEGF) [205,206,207,208] in an interaction that induces and supports angiogenesis.

All the pro-tumor properties of *CXCL1* are confirmed by clinical tumor studies, which showed that *CXCL1* expression increased with cancer progression. This was also shown in studies of patients with cancers such as bladder cancer [189], colorectal cancer [209], gastric cancer [210], hepatocellular carcinoma [114], laryngeal squamous cell carcinoma [211], prostate cancer [212] and renal cell carcinoma [213].

The high importance of *CXCL1* in tumorigenesis allows for the development of effective therapies targeting this chemokine. An example of this is HL2401—an anti-*CXCL1* monoclonal antibody, which shows promising results against bladder cancer and prostate cancer in animal experiments [214]. Additionally, CXCR2 inhibitors such as SB225002 are currently being tested [215,216,217]. This compound has anti-tumor effects on various cancer cell lines as well as animal models. Another possibility is the use of dual CXCR1/CXCR2 inhibitors such as repertaxin [218], ladarixin [219], SCH-479833 or SCH-527123 [220]. They inhibit not only *CXCL1* activity but also other CXCR2 agonists.

Anti-cancer therapies do not just have to be associated with a reduction in the expression and effects of *CXCL1*. For example, as T cells do not express CXCR2 [221,222], if the expression of the receptor on these cells was increased, then such T cells would be specifically recruited to sites with a high concentration of CXCR2 ligands, including *CXCL1*. Therefore, such modified T cells could be recruited to solid tumors with an elevated concentration of *CXCL1*, which would result in the initiation of an anti-tumor immune response [221,222]. This process can be exploited in immunotherapy, e.g., using modified autologous chimeric antigen receptor (CAR)-T cells.

## 8. Perspective for Further Research

The regulation of *CXCL1* expression and activity is very well researched. In particular, we already know much about the regulation of *CXCL1* gene transcription and the role of miRNAs in *CXCL1* mRNA stability. However, there are areas of knowledge that are poorly understood and should be further investigated in the near future. In particular, little is known about the following aspects of the regulation of *CXCL1* expression and activity:-The regulation of *CXCL1* mRNA stability: The effect of IL-17 on this process is fairly well established, but the role of IL-1 and TNF-α requires further investigation. Additionally, the exact mechanism causing the low stability of *CXCL1* mRNA in cells not stimulated by any cytokines are unknown;-The mechanisms of sorting the *CXCL1* to vesicles and its release outside the cell: *CXCL1* is an important component of cellular responses to dangerous agents. For this reason, intercellular signaling involving *CXCL1* must sometimes be very rapid, and in some cases, the release of *CXCL1* from the cell must be immediate. However, very little research is devoted to the regulation of *CXCL1* levels outside cells;-Proteolytic processing: Until now, not all proteases involved in proteolytic processing of *CXCL1* have been identified. The investigation of this mechanism could indicate new therapeutic avenues for diseases in which *CXCL1* plays an important role;-*CXCL1* heterodimerization with other CXC chemokines and interactions between CXC chemokines: A characteristic feature of CXC subfamily chemokines is the eight different CXC chemokines that cause the same ability to activate CXCR2 at similar concentrations. Evolutionary pressures caused many duplications of the ancestral gene of all ELR^+^ CXC chemokines. However, it is not entirely clear why, perhaps due to the different regulation of expression for each chemokine in this group. This argument can be linked to the heterodimerization of ELR^+^ CXC chemokines. With 8 ELR^+^ CXC chemokines, we have 28 different heterodimers, 8 homodimers and 8 monomers—a total of 44 different molecules that can activate CXCR2 in slightly different ways. This is almost 5.5 times more than the number of chemokines. The interaction with GAG is also important in this model.

## Figures and Tables

**Figure 1 ijms-23-00792-f001:**
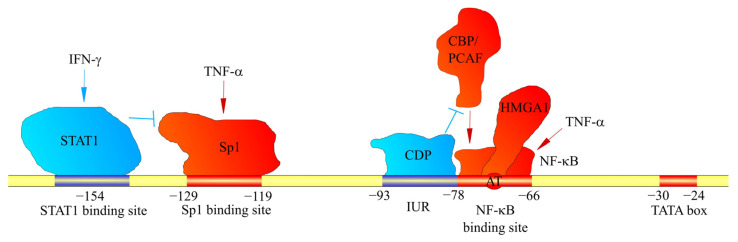
Factors affecting *CXCL1* gene transcription. The *CXCL1* promoter contains binding sites for factors that bind to these sites. They either increase (shown in red) or decrease (shown in blue) *CXCL1* expression. In particular, the *CXCL1* promoter contains a binding site for STAT1 and Sp1. STAT1 and Sp1 activated by IFN-γ and TNF-α, respectively, attach to these sites. Sp1 increases *CXCL1* expression; however, STAT1 inhibits Sp1 binding to the *CXCL1* promoter. Another mechanism regulating *CXCL1* expression is IUR, a region that directly borders the NF-κB binding site. CDP binds to IUR, which inhibits the recruitment of CBP and PCAF coactivators by NF-κB. This prevents the induction of *CXCL1* expression by NF-κB.

**Figure 2 ijms-23-00792-f002:**
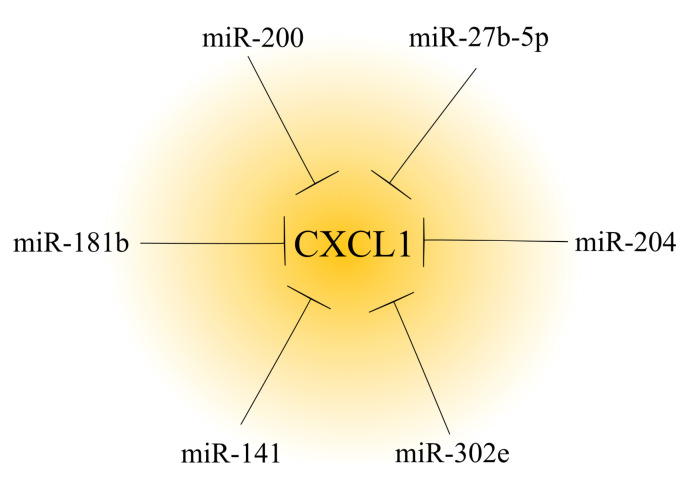
Effect of microRNAs on *CXCL1* expression in a tumor. In cancer tumors, *CXCL1* expression is regulated by microRNAs. Based on the available literature, six different microRNAs that decrease *CXCL1* expression have been identified to date.

**Figure 3 ijms-23-00792-f003:**
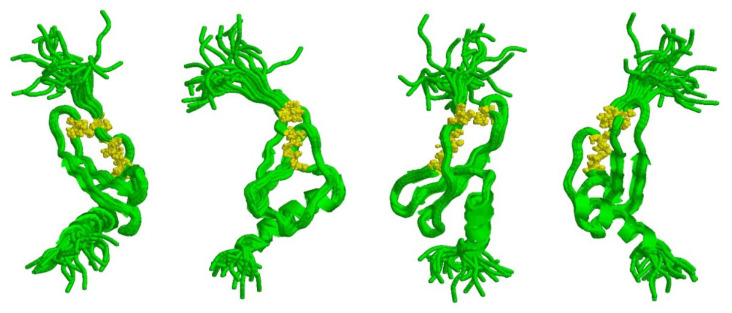
Structure of *CXCL1*. The tertiary structure of *CXCL1* with the two disulfide bridges is highlighted in yellow. The *N*-terminal structure of *CXCL1* is not stabilized; therefore, the figure shows many different possible conformations of this, as well as the other terminus of the *CXCL1* chain. The graphic was created using RasMol 2.7.4.2 [136,137], and the *CXCL1* structure was deposited in the Protein Data Bank (PDB) [138] under the identifier 1MGS [133,134].

**Table 1 ijms-23-00792-t001:** Proteins that bind to the *CXCL1* gene promoter, leading to a change in its expression.

Name of the Factor	BINDING SITE	Effect on Expression	Notes	References
p50:p65 NF-κB	−78 bp to −66bp	↑	High basal NF-κB activity in cancer conditions; high basal *CXCL1* expression in tumors. Activated in inflammation	[33,34,35,36]
p50:p50 NF-κB	?	↓	Prevention of chronic liver disease	[49]
HMGA1	From −74 bp to −73 bp	↑	Essential in the full activation of the *CXCL1* promoter by NF-κB	[34]
CDP	from −93 bp to −78 bp	↓	Reduction in *CXCL1* expression by disruption of NF-κB function	[50,52,53]
PARP1	from −93 bp to −78 bp	↓	PARP1 binding in the inactive state. Inhibition of NF-κB binding to the *CXCL1* promoter	[55,56]
CUX1	−94 bp to −84 bp	↑	Enhancement of *CXCL1* expression by the joint action of IL-17 and TNF-α	[51]
Sp1	−129 bp to −119 bp	↑	Significant in basal *CXCL1* expression and in upregulation of *CXCL1* expression by IL-17 or TNF-α	[34,50]
STAT1	−154 bp	↓	Reduction in *CXCL1* expression by IFN-γ through disruption of Sp1 function	[57]
STAT1/STAT4	?	↑	Enhancement of *CXCL1* expression by IL-35	[58]
HIF-1 and HIF-2	?	(↑)	Increased expression of *CXCL1* in hypoxia. No precise studies on the direct effect	[84]
MEIS1	−277 bp	(↑)	Sequence identified as potential binding site but non-functional. Factor influence indirect. Relevant in cancer, particularly in ovarian cancer	[90]
Erg-1	−367 bp and −134 bp	↑	Important in cancer, especially in esophageal cancer	[88]
MITF	−375 bp	↑	Important in cancer, especially in melanoma cancer	[91]
Snail	from −984 bp to −301 bp	↑	Increased *CXCL1* expression during EMT, important in cancer during metastasis formation	[94]
SMAD4	−1247 bp and −560 bp	(↓)	Sequences identified as potential binding sites but non-functional. Theoretically, when TGF-β action is reduced, the effect of SMAD4 is abolished, and thus *CXCL1* expression increases	[61]
SETD2	from −2.0 to −1.5 kbp	↓	This is the enzyme that causes histone methylation. The exact mechanisms of how epigenetic changes in this region affect *CXCL1* expression are not known	[101]
HeyL	−2 kbp	↑	Notch signaling element. Relevant for cancer	[97,98]
Mutated p53	?	↑	Relevant in cancers with *TP53* gene mutation	[74,75]
p63	−3 kb	↑	Relevant in cancer, especially in pancreatic ductal adenocarcinoma cells	[77]
MAFF	−15 kpb, −12.5 kpb and −7.5 kbp	↑	Induction of *CXCL1* expression in human term myometrium, immediately before birth. The exact functions of *CXCL1* in labor are unknown	[99]

↑—a factor that increases the expression of *CXCL1*; ↓—a factor that reduces the expression of *CXCL1*; (↓)—A factor that reduces *CXCL1* expression with an identified direct binding site in the *CXCL1* promoter, but the effect of the indicated factor is only indirect.

## Data Availability

Not applicable.

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
