# Peer review of "CXCL1: Gene, Promoter, Regulation of Expression, mRNA Stability, Regulation of Activity in the Intercellular Space"

_ijms, 2022, doi:10.3390/ijms23020792_

Round 1

Reviewer 1 Report

In this review, Jan Korbecki and collaborators analysed the mechanisms of CXCL1 expression regulation in cancer. They focused the attention on the regulation of CXCL1 expression through the regulation of CXCL1 translation and mRNA stability, including the involvement of transcription factors (NF-kB, p53), the effect of miRNAs and cytokines (including IFN-g, IL-1b, IL-17, TGF-β and TNF-a). The mechanisms regulating CXCL1 activity in the extracellular space, including proteolytic processing, CXCL1 dimerization and the influence of the ACKR1/DARC receptor on CXCL1 localization was also described. In addition, the role of CXCL1 in cancer and its clinical significance is discussed.

Overall, this article is of interest.

Specific comments/questions:

  • References should be updated. For instance, the role of CXCL1 in inflammation-driven skin carcinogenesis model (Finegan et al 2015, doi: 1158/0008-5472.CAN-13-3043) should be included and discussed.
  • The quality of the English writing requires a substantial improvement

Author Response

Review1

Comments and Suggestions for Authors. In this review, Jan Korbecki and collaborators analysed the mechanisms of CXCL1 expression regulation in cancer. They focused the attention on the regulation of CXCL1 expression through the regulation of CXCL1 translation and mRNA stability, including the involvement of transcription factors (NF-kB, p53), the effect of miRNAs and cytokines (including IFN-g, IL-1b, IL-17, TGF-β and TNF-a). The mechanisms regulating CXCL1 activity in the extracellular space, including proteolytic processing, CXCL1 dimerization and the influence of the ACKR1/DARC receptor on CXCL1 localization was also described. In addition, the role of CXCL1 in cancer and its clinical significance is discussed. Overall, this article is of interest.

Specific comments/questions:

  • References should be updated. For instance, the role of CXCL1 in inflammation-driven skin carcinogenesis model (Finegan et al 2015, doi: 1158/0008-5472.CAN-13-3043) should be included and discussed.
  • The quality of the English writing requires a substantial improvement

According to Reviewer remark we corrected and supplemented references and the manuscript has been checked by a native speaker. We included the certificate.

Reviewer 2 Report

This article is comprehensive and potentially of great interest to a wide audience. The limitations for this review are listed below:

  1. The quality of Figure 3 must be improved. It is difficult to see the image in its current form.
  2. There are two section "7" in the manuscript("Significance of CXCL1 in tumors" and "Perspective for future research"). Please re-number them properly.

Author Response

Comments and Suggestions for Authors

This article is comprehensive and potentially of great interest to a wide audience. The limitations for this review are listed below:

  1. The quality of Figure 3 must be improved. It is difficult to see the image in its current form.
  2. There are two section "7" in the manuscript("Significance of CXCL1 in tumors" and "Perspective for future research"). Please re-number them properly.

According to Reviewer remark we corrected Fig.3 and the numbers of sections in whole manuscript.